# The Diagnosis of Perineural Invasion: A Crucial Factor in Novel Algorithm of Coexistence of Conventional and Nerve-Sparing Radical Hysterectomy

**DOI:** 10.3390/diagnostics11081308

**Published:** 2021-07-21

**Authors:** Andrzej Skręt, Joanna Ewa Skręt-Magierło, Mariusz Książek, Bogusław Gawlik, Joanna Bielatowicz, Edyta Barnaś

**Affiliations:** 1Health Care Center, Department of Obstetrics and Gynecology with Oncological Gynecology, Krakowska 91, 39-200 Dębica, Poland; askret42@gmail.com (A.S.); b.gawlik1@interia.pl (B.G.); jbielatowicz@poczta.fm (J.B.); 2College of Medical Sciences, University of Rzeszow, Rejtana 16c, 35-959 Rzeszów, Poland; ebarnas@interia.eu; 3Clinical Department of Morphological Sciences, Clinical Provincial Hospital No. 2 St. Jadwiga Krolowej in Rzeszow, Lwowska 60, 35-301 Rzeszów, Poland; mariobwp@pro.onet.pl

**Keywords:** perineural invasion, diagnosis, cervical cancer

## Abstract

Nerve-sparing radical hysterectomy (NSRH) was introduced to mitigate adverse effects associated with conventional radical hysterectomy (CRH) in cervical cancer. However, the introduction of NSRH was compromised by possible existence of perineural invasion (PNI). Additionally, the coexistence of NSRH and CRH is currently the fact. The aim of the study was to review the literature and attempt to construct a novel and preliminary PNI diagnostic algorithm that would establish the coexistence of NSRH and CRH in one system of early-stage cervical cancer (ESCC) surgical treatment. This algorithm takes into account the PNI risk factors and current and future diagnostic methods such as imaging and biopsy.

## 1. Introduction

The surgical methods for cervical cancer treatment have been developing over the last 120 years. The introduction of each new surgical technique resulted in a new compromise that needed to be resolved. In the Werheim era, at the beginning of radical hysterectomy (RH), the first compromise to be worked out was a radical technique connected with high mortality and cervical cancer, which used to be a fatal disease. The compromise was resolved by reduction of mortality due to progress in anesthesiology and transfusiology and by the introduction of safer vaginal routes of RH. Additionally, RH classifications were developed that gave the base to adjust the extent of dissection to stage cervical cancer (CC). Other side effects connected with RH were obvious infertility and high morbidity connected with wide lymph nodes dissection. These side effects of RH were partially diminished by the introduction of radical trachelectomy and the sentinel node technique. Other compromises were connected with side effects due to surgical trauma to the pelvic autonomic nervous system. The introduction of the nerve-sparing radical hysterectomy (NSRH) technique was a remedy for this issue. However, it soon occurred that perineural invasion (PNI) existed in CC as in other malignancies. This phenomenon created a new compromise between the application of NSRH and the existence of PNI. Some authors disregard it and claim that NSRH should be the standard of care in CC surgery [1], while others state that since CC exhibits a tendency to PNI and is correlated with poor prognosis, the recommendations and guidelines for conducting NSRH need to be defined [2]. It means that coexistence of NSRH and conventional radical hysterectomy (CRH) is possible, and thus a system of coexistence is needed. In this system diagnosis of PNI presence has a pivotal role because it determines the choice of CRH. Unfortunately, preoperative diagnosis of PNI is difficult and eligibility criteria do not exist. Preoperative PNI diagnosis was crucial in the choice between NSRH and CRH application. Taking such a decision without knowledge about PNI status may have some negative effects. The data from four control randomized trials (CRT) on efficacy of NSRH vs CRH in early-stage cervical cancer (ESCC) were analyzed. A total of 211 patients were randomized to NSRH (105 patients) or CRH (106 patients) [3,4,5,6]. Zhu et al. found that the incidence of PNI in ESCC in eight reports amounted from 7% to 35.1% [2]. Having a closer look at the calculation of an average value of 16% in Wei et al., it appeared that in the group subjected to NSRH, 17 patients had PNI left in spared nerves, while in the CRH group, 89 out of 106 had dissected uninvolved nerves [7]. The above incorrect decisions were an argument for preoperative or intraoperative PNI diagnostics.

## 2. The Purpose

The aim of the study was to review the literature to find out the data that would allow us to develop a PNI diagnostic algorithm that would establish the coexistence of NSRH and CRH in one system of ESCC surgical treatment.

## 3. The Data Dealing with PNI in CC

### 3.1. Pelvic Nerve Involvement in Cervical Cancer and Other Malignancies

Traditionally, PNI in patients suffering from CC is diagnosed based on the postoperative specimen after conventional RH and is not useful in preoperative therapy guidelines. It enables only identification of the clinical data that are risk factors for PNI. It is worth noting that authors assessed PNI in the cervix and/or parametria and identified the place of these findings.

It was found that the transfer mode of PNI was continuous, non-jumping, and direct spreading. Thus, tumor cells migrate through a “neural road” [2]. It supports the idea of creating classification of PNI localizations, e.g.,: PNI in cervix—class I; uterine branches—class II; inferior hypogastric plexus—class III; and pelvic splanchnic nerve and/or inferior hypogastric nerve and its vesical and anorectal branches—class IV.

Provisional classification suggested by us is a novel idea. Miller et al. suggested a simple two-class PNI classification in head and neck carcinoma divided into intramural and extramural. The authors determined the prognostic significance of this classification to clinical outcome because that may help to define a cohort of patients that may require more aggressive management. Additionally, they measured the distance of each PNI focus to the tumor edge and the size of the largest nerve involved [8].

Table 1 presents an overview of the studies concerning the incidence and the most important parameters of PNI in CC.

The material from the reports in Table 1 is non-homogenous and differs in respect to clinical stage and percentage of PNI existence. The authors of these reports presented clinical and histopathological features that are more frequent in patients with PNI and are de facto risk factors of PNI.

The most common risk factors include: FIGO stage, tumor size, depth of stromal invasion, and parametrial involvement.

### 3.2. NSRH

RH in CC treatment was introduced by Wertheim a decade after Halsted’s radical mastectomy invention. It sacrificed the integrity of the pelvic nervous anatomic system on the altar of radicality. In 1988, Sakamoto presented a method of RH that improved this operation to prevent urological complications. It preserves the pelvic anatomic nerve bundle under cardinal ligament. According to Kato et al., the cardinal ligament is composed of two parts: the vascular part containing the uterine artery and vein and the vaginal and vesical artery; and the neural part containing autonomic nerve fibers, hypogastric nerve, and internal hypogastric plexus [21]. Dividing the cardinal ligament and dissecting only its vascular part was the first nerve sparing technique of RH. Currently, the last-mentioned technique is being revived as nerve plane-sparing RH—a simplified technique of NSRH [22].

However, the current standard techniques of NSRH visualize and spare separately the essential pieces of the pelvic nervous system such as hypogastric nerves, pelvic splanchnic nerves, and the inferior hypogastric plexus and its vesical branches.

According to Sakuragi et al., the nerve-sparing procedure can be combined with various types of RH and fulfills the criteria of precision cancer surgery, which has a pivotal role in personalized treatment [23]. It maximizes the oncological outcome while minimizing the deterioration of a patient’s quality of life (QoL).

### 3.3. PNI Diagnostic Methods

#### 3.3.1. Imaging Methods

The imaging methods are not accepted for formal staging purposes in patients with cervical cancer. It was explained that FIGO staging is intended only for comparison purposes. However, it was stressed that findings of CT, MRI, and fusion images such as PET/CT may be used to guide treatment options [24].

In 2018, FIGO published a new staging system that, after other institutions’ suggestions, integrated dual pretreatment staging by FIGO and TNM. Obviously, some data assessed by TNM and FIGO such as metastatic lymph nodes (FIGO 3C), hydronephrosis (FIGO 3B), and distant metastases (FIGO 4B) may be diagnosed with imaging techniques [25].

According to Capek, modern methods with high resolution techniques are candidates to be elements of non-invasive PNI diagnostics. He examined their role in a group of patients with pelvic malignances—prostate, vesical, anal, and cervical cancers—and found that all these neoplasms are spreading in a similar manner inside the lumbosacral plexus, giving the picture of lumbosacral plexopathy. The latter also may be caused by many other diseases, such as diabetes mellitus and previous pelvic injuries. Capek et al. report the cases of PNI visualization with the use of MRI, FDG/PET CT, and choline PET/CT. This presentation is not complete with respect to the usefulness of the above methods’ assessment specificity and sensitivity, but it shows new ways of non-invasive PNI diagnosis. Capek et al. reported some findings in imaging methods that may be indicative of PNI. According to them, the affected nerves are typically enlarged on T1-weighted sequences, often with irregular and nodular contours; on T2-weighted images, they are hyperintense, but the T2 signal might extend beyond tumor infiltration due to “downstream” effects [26].

#### 3.3.2. Biopsy

Biopsy followed by microscopic examination is the only unequivocal diagnostic preoperative method of PNI confirmation. It was previously applied in prostatic or pancreatic neoplasms. In cervical cancer staging, such biopsy still awaits its application. Magierło-Skręt et al. suggested the use of uterine branches of inferior hypogastric plexus biopsy intraoperatively, accompanied by frozen section examination [11]. The presence or absence of PNI in such a specimen would be decisive in choosing between NSRH and CRH. According to Zhu et al., this approach might be ambiguous when trying to assess the status of the nerve stem, with additional open questions: where should the examination be performed, and how many sites should be examined? Additionally, Zhu et al. questioned the accuracy of it and suggested simultaneous identification of HPV16 E6 and S100 by double immunofluorescence [2], which, however, in our opinion, is not suitable in the frozen section setting due to the lengthy immunofluorescence procedure. Taking this into account, we suggest performing a laparoscopic biopsy a few days before the RH. This approach would allow the use of routine hematoxylin and eosin staining on formalin-fixed, paraffin-embedded tissue, which provides much more reliable results, supported by immunohistochemistry if needed. Such a procedure, performed on the right pelvis side, is shown in the Figure 1 and consists of the following steps: opening of retroperitoneal space above the right ureter, replacing it laterally to visualize the hypogastric nerve, releasing and mobilizing to the level of the right inferior hypogastric plexus. Then, the area located medially to the right inferior hypogastric plexus, corresponding with its right uterine branch, is biopsied.

The microphotographs of biopsied structures (Figure 2) exhibit small nerve fibers surrounded by adipose tissue (Figure 2A—black arrows and Figure 2C), highlighted by immunohistochemistry S100 staining (Figure 2B,D). The above data may support the idea that pre-surgery laparoscopic biopsy of the uterine nerve branches may have high prognostic value in making preoperative decisions. The above-described procedure is not yet supported by robust study and requires further research.

We realize that during the above-described random biopsy, only a small part the uterine branch is taken and evaluated. It may make the biopsy non-representative in the assessment of PNI status. Note that the uterine branch of the inferior hypogastric plexus is nervous tissue that can be sacrificed because parallel hysterectomy permits it. Second, biopsies of other pelvic nervous system are impossible, although it would permit assessment of the full range of PNI.

### 3.4. Cancer Nerve Crosstalk and Diagnostic/Therapeutic Methods Based on It

Some identified mechanisms involved in PNI may be useful in preoperative diagnosis and treatment of PNI.

The sympathetic nervous system regulates the tumor microenvironment, and the cervix and uterus are innervated by the autonomic nervous system (ANS) [27]. It has been proved that activation of the ANS, especially the sympathetic division in particular, modulates gene expression that promotes metastasis of solid tumors [28]. Recent research showed that Schwann cells have a unique and specific affinity for tumors and that they aid in tumor dissociation, migration, and invasion [29,30,31].

Neural cell adhesion molecule 1 (NCAM1) is an important molecular mediator of Schwann-cell-directed PNI [32,33]. Other researchers have focused on the genetic mechanism of PNI, including a gene defect and the role of tumor suppressor gene (p. 73) [34]. Additionally, other recent studies confirmed the complex information between nerves and the tumor cells invading the nerves, where there has been observed signal transduction through neurotrophic growth factors, such as neurotrophin and granulocyte colony-stimulating factor (G-CSF) [35,36].

Another of them are the chemokines that are involved in the process of PNI as one of pivotal factors [37].

Generally, chemokines promote the progression of cancer, accelerating tumor growth by activation of growth factor receptors. Chemokines are a group of small peptides secreted by various cell types, including certain tumor cells [38,39,40]. Currently, 50 detected chemokines are divided into four subtypes: C, CC, CXC, and CX3C [37].

The axis of chemokine CCL-2 and its receptor CCR2 are identified as the elements in cancer–nerve crosstalk, forming a tumor microenvironment that facilitates the PNI. CCL2 is mainly secreted by Schwann cells that first arrive at the sites of cancer cells and bind to the CCR2, promoting proliferation, migration, invasion, and epithelial–mesenchymal transition (EMT). In turn, tumor cells trigger expression of metalloproteins in Schwann cells, which dissolve matrix [37]. That way the cancer–nerve crosstalk forms the tumor microenvironment (TME) and facilitates PNI. Serum CCL2 is elevated in patients with PNI, and it may in the future be a potential marker of PNI.

## 4. Oncological Outcome in Cervical Cancer Patients with PNI

PNI is detectable in several malignancies, such as cancers of the pancreas, prostate, head and neck, as well as colon. This feature correlates with unfavorable clinical outcomes. The postoperative assessment of specimens from RH of CC revealed the incidence of PNI in 7.0% to 35.1% (Table 1). Oncological outcome was measured in several reports using disease-free survival (DFS) and overall survival (OS). Some authors found no connection between PNI and oncological outcome in reports based on the clinical data of patients with non-homogenous parameters [11,14,15,19]. However, others confirmed that PNI was related to DFS and OS in univariate but not in multivariate analysis [2]. The latter was explained by the strong correlation between PNI and other poor prognosis factors, and some identified the existence of PNI as an independent risk factor both in univariate and multivariate analyses of poor oncological outcome [12]. It is worth noting that all of the above-mentioned reports studied the existence of PNI in postoperative specimens only and allowed to choose adjuvant therapy. On the contrary, only preoperative PNI diagnosis may permit selecting the method of primary treatment.

## 5. The PNI Diagnostic Algorithm in ESCC Patients Qualified for Surgical Treatment with RH

The suggested algorithm of CRH and NSRH is presented in Figure 3. The rationale for it is the result of several reasons: both these methods have pros and cons. NSRH improves quality of life, while CRH gives better results with a lower risk of recurrence but with reduced quality of life. NSRH carries the risk of remaining possible PNI, which is correlated with poor oncological outcomes. The survival after ESCC is high (5-year OS is 85%), and many patients treated with RH are long term survivors and may experience better QoL after NSRH [41,42,43].

The proposed algorithm is an attempt at a rational balance between NSRH and CRH in terms of quality of life and oncological outcomes (Figure 3).

The first selection of patients with ESCC would be those with PNI risk factors based on the results of histopathological examination of postoperative specimens after RH.

The preoperative and postoperative histological assessment of PNI requires not only H&E staining but also immunohistochemical examination. It is necessary to establish a risk model exploring the relationship between nerve–tumor distance and nerve diameter with clinical outcomes [44].

Among these factors, the first is the clinical stage. A discriminatory degree of clinical stage would be FIGO stage IIA. According to Kato et al., for FIGO IIA_1_ < 2 cm with a small lesion in the vagina and IA_2_ and IB_2_ < 2 cm, patients could be operated on with a unilateral NSRH. However, according to this author, the above-mentioned cases of unilateral NSRH are associated with poorer oncological outcomes [21].

Other risk factors representing discriminatory values are tumor size > 4 cm, deep cervical stromal invasion > 2/3, and parametrial invasion. These factors can be evaluated using the imaging methods permitted by the new FIGO classification.

Thus, an indication for NSRH would be cases where tumor < 4 cm, invasion < 2/3, and no infiltration of the parametria.

A special group are cases with the stage of IB_2_ and IIA_1_ advancement in which neoadjuvant chemotherapy NACT was used. According to Zhu et al., these patients should be treated with CRH because NACT may mask the presence of PNI [2].

If no risk factor is present, NSRH can be used. When all these factors are present, the patient qualifies for CRH. When there are single factors, the patient may undergo diagnostic methods such as high-resolution imaging methods or preoperative biopsy and/or intraoperative biopsy.

Imaging methods are not conclusive and may only produce PNI suspicion. In that case, there is an indication to preoperative and intraoperative biopsy. It is worth noting that the imaging methods and biopsies are currently experimental and they lack standardization. Imaging methods and biopsy do not exhaust all the future possibilities of PNI diagnosis.

It is worth noting the potential application of a method based on the virus NV 1066 as an intraoperative diagnostic [45]. Another method includes an assessment of the expression of the CCL2 chemokine in the serum.

The algorithm also takes into account the management of PNI in the postoperative specimen. In this case, adjuvant treatment needs to be applied.

## 6. Limitations

The aim of the paper was to establish a proposal for a current and future system of coexistence of CRH and NSRH that would be able to increase QoL together with improved oncological outcome. However, this idea carries some limitations of its effectiveness. First of all, the value of PNI risk factors has not been precisely established. Second, the PNI predictive value of diagnostic methods such as imaging and biopsy has still not been determined. Additional studies are required to resolve the above-mentioned two limitations.

## 7. Conclusions

This paper presents the novel and preliminary idea of a diagnostic/therapeutic system governing the coexistence of CRH and NSRH in surgery of CC. In this system, current and future methods of PNI identification were presented that would permit a potent selection of CRH and NSRH methods. The paper does not attempt to present a solution but presents a verified path in the usual never-ending story of diagnostic and therapeutic methods development.

## Figures and Tables

**Figure 1 diagnostics-11-01308-f001:**
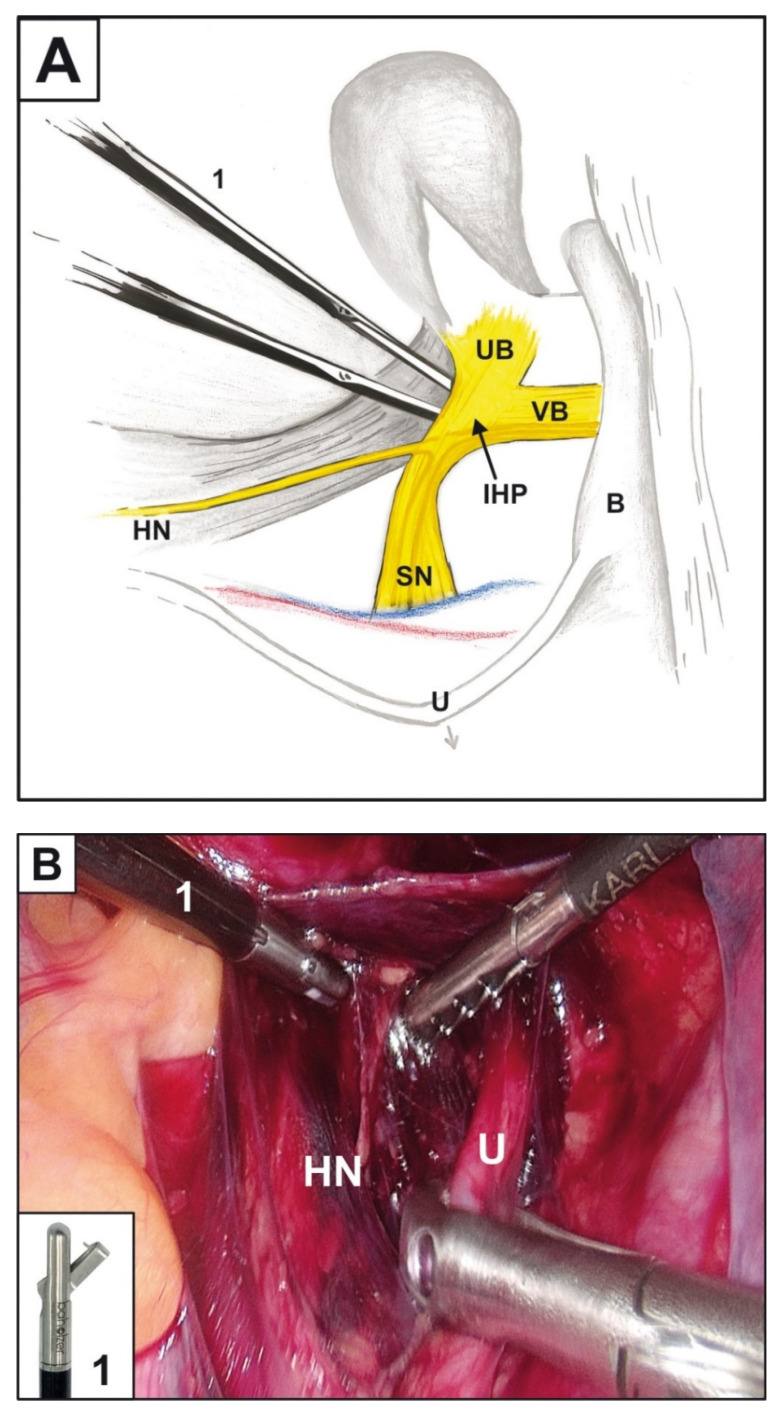
Laparoscopic biopsy of uterine branches: (**A**). Scheme; (**B**). Laparoscopic view. Legends: HN—hypogastric nerve; UB—uterine branch; VB—bladder branch; IHP—inferior hypogastric plexus; SN—splanchnic nerve; U—ureter; B—bladder; 1—biopsy punch with thorn.

**Figure 2 diagnostics-11-01308-f002:**
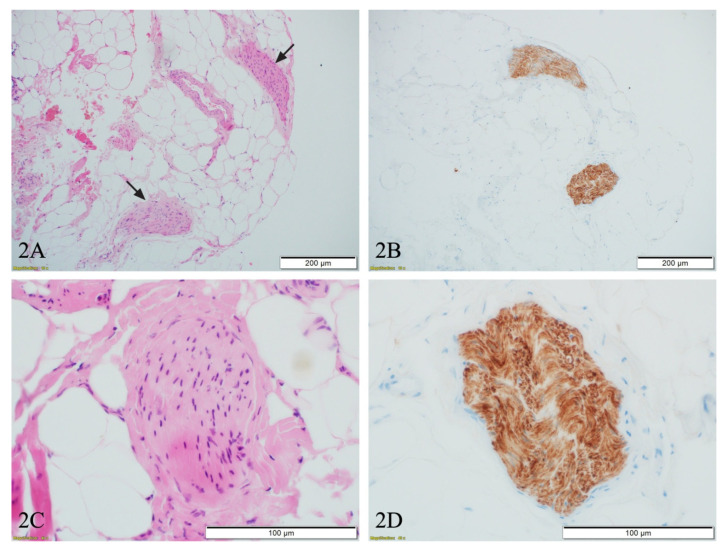
The tissue from laparoscopic biopsy. (**A**): Two nerve fibers (black arrows) surrounded by adipose tissue 1B, H&E, objective ×10; (**B**): Corresponding nerves highlighted with S100 immunohistochemical staining, objective ×10; (**C**): High magnification of nerve fiber, showing no evidence of perineural invasion, H&E, objective ×40; (**D**): Corresponding nerve fiber highlighted with S100 immunohistochemical staining, objective ×40.

**Figure 3 diagnostics-11-01308-f003:**
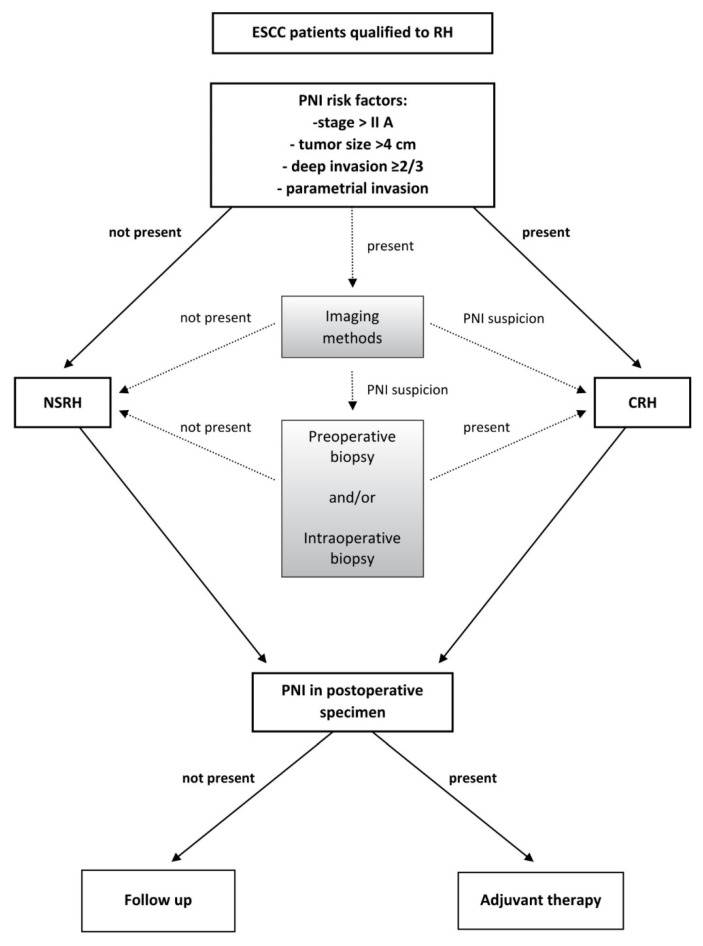
The PNI diagnostic algorithm in ESCC patients qualified to surgical treatment with RH. The shadow areas are suggested methods applicable in future.

**Table 1 diagnostics-11-01308-t001:** The overview of the studies on PNI in CC.

Authors, Years, References	Samples/ with PNI (N)	Localization of PNI	Frequency of PNI (%)	Stage	Grade	Tumor Size with PNI	Depth of Invasion with PNI
Wei et al., 2016 [7]	206/33	Cervix *n* = 28, Cervix and parametrium *n* = 5	16	IB1-B	G1-3	<4 cm *n* = 16 >4 cm *n* = 17	<1/2 *n* = 6 >1/2 *n* = 27
Zhu et al., 2018 [9]	201/18	Cervix *n* = 15 Cervix and parametrium *n* = 3	8.57	IA2-IIA2	N/A	<4 cm *n* = 5 >4 cm *n* = 13	<2/3 *n* = 2 >2/3 *n* = 16
Tang et al., 2019 [10]	406/43	Cervix *n* = 43	10.59	IA2-IIA1	N/A	<4 cm *n* = 27 >4 cm *n* = 16	<2/3 *n* = 6 >2/3 *n* = 37
Skręt-Magierło et al., 2014 [11]	50/9	Parametria *n* = 9	18	IB1-IIB	G1-G3	<4 cm *n* = 3 >4 cm *n* = 6	<15 mm *n* = 1 >15 mm *n* = 8
Horn et al., 2010 [12]	194/68	Cervix	35.1	IB1-IIB	G1-G3	N/A	N/A
Meinel et al., 2011 [13]	194/68	cervix	35.1	IB1-IIB	G1-G3	N/A	<66% *n* = 67 >66% *n* = 126
Elshawi et al., 2011 [14]	190/24	Cervical stroma *n* = 24	12.5	IB1-IIA	G1-G3	>4 cm *n* = 10 <4 cm *n* = 14	>2/3 *n* = 18 <2/3 *n* = 6
Cho et al., 2013 [15]	185/13	Cervix *n* =13 Parametrium *n* = 8	7	IA2-IIA2	N/A	<4 cm *n* = 6 >4 cm *n* = 7	<1/3 *n* = 2 >1/3 *n* = 11
Baiocchi et al., 2017 [16]	345/16	Cervix *n* = 42 Parametrium *n* = 16	4.6	IA2-IB2	G1-G3	N/A	N/A
Vural et al., 2017 [17]	111/ 34	Parametrium *n* = 26		IIA-IIB	N/A	<4 cm *n* = 15 >4 cm *n* = 19	<2/3 *n* = 1 >2/3 *n* = 33
Memarzadeh et al., 2003 [18]	93	Parametrium	7.5	IA2-IIA	N/A	N/A	N/A
Tavares et al., 2009 [19]	301	Cervix	27	I-II	N/A	N/A	N/A
Ozan et al., 2008 [20]	36	N/A	33	IB1-IB2	N/A	N/A	N/A

N/A: not available.

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
