# Peer review of "The Diagnosis of Perineural Invasion: A Crucial Factor in Novel Algorithm of Coexistence of Conventional and Nerve-Sparing Radical Hysterectomy"

_diagnostics, 2021, doi:10.3390/diagnostics11081308_

Round 1
Reviewer 1 Report
Comments on Diagnostics manuscript (diagnostics-1291368): The diagnosis of perineural invasion: a crucial factor in novel algorithm of coexistence of conventional and nerve sparing radical hysterectomy
This paper aims to review the literature and develop a perineural invasion (PNI) diagnostic algorithm. It is potentially novel; however, some major issues should be addressed by the authors:
- First, PNI of cervical cancer is associated with LVSI and lymph node metastasis and can be used as an index for the determination of post‑operative radiotherapy for cervical cancer patients. PNI itself is not an independent prognostic factor and is currently not stated in the treatment guidelines. Moreover, PNI is associated with the depth of invasion, tumor size and more advanced stage of the disease in early-stage cervical cancer with high-risk of recurrence. No concrete association with a worse prognosis was observed. Therefore, a preoperative diagnosis of PNI might not be necessary.
- Second, a nerve-sparing radical hysterectomy (NSRH) is recommended for selected cases under the condition that the NSRH does not impair the radicality of surgery. The reported incidence of the perineural invasion in cervical cancer ranges from 7.5% to 35.1%. The authors should discuss more about the role of magnetic resonance image (MRI) and positron emission tomography/computed tomography (PET/CT) in assessing the perineural spread in cervical cancer (sensitivity and specificity) preoperatively. Personally, I do not believe that the aforementioned image tools could provide the aforementioned information, although if the authors who would like to conduct it, it is highly appreciated. Therefore, the authors should conduct patient-based data analysis to clarify the sensitivity and specificity of MRI or PET/CT on perineural invasion.
- The authors suggested performing the laparoscopic biopsy a few days before the RH was crucial in the choice between NSRH and conventional radical hysterectomy (CRH) application. Is it practical in routine clinical practice? Even though laparoscopic approach can be considered as a less invasive procedure and in addition, if the authors recommended feasibility of preoperative laparoscopy, why the authors did not recommend the intraoperative frozen pathology for this purpose? Less invasive surgery was not equal to fewer complications. Therefore, the authors can introduce to what we should do to minimize the risk of nerve fibers damage during pre-operative laparoscopic biopsy.
- The biopsy is performed on the right side, please specify the reason. Although cone was not recommended before the definite surgery for cervical cancers (CRH or NSRH, if the PNI is strongly positive with deep stromal invasion (> 2/3), MRI or PET/CT might be a diagnostic value. However, cone may directly provide the more confirmation of the deep stromal invasion.
- The strong limitation of this study is that there is no mention of PNI in NCCN guideline and the authors should conduct the original analysis on this future system.
- Figure 1 Ligand: U indicates ureter instead of urethra.
- Figure 3: The PNI diagnostic algorithm is not clear. It should clearly indicate who will be beneficial for preoperative biopsy.
- Lastly, in line 1, the type of this article is mentioned as “perspective”. We remine the editors that perspective is not one type of articles in diagnostics. We cannot distinguish what kind of this article should be included; however, it is not the original article. The manuscripts submitted to Diagnostics does not fit into any of the following categories:
- Articles: Original research manuscripts. The journal considers all original research manuscripts provided that the work reports scientifically sound experiments and provides a substantial amount of new information. Authors should not unnecessarily divide their work into several related manuscripts, although Short Communications of preliminary, but significant, results will be considered. The quality and impact of the study will be considered during peer review.
- Reviews: These provide concise and precise updates on the latest progress made in a given area of research. Systematic reviews should follow the PRISMA guidelines.
- Case reports: Case reports present detailed information on the symptoms, signs, diagnosis, treatment (including all types of interventions), and outcomes of an individual patient. Case reports usually describe new or uncommon conditions that serve to enhance medical care or highlight diagnostic approaches.
- Guidelines: These papers cover step-by-step systematical procedural instructions to clinicians for the care of patients with specific conditions. They can be consensus-based or clinical practice guidelines.
- Protocols: Protocol articles present a detailed introduction for proposed or ongoing clinical research, outlining the hypothesis, related rationale and methodology.
- Interesting Images: Diagnostics encourages the submission of Interesting Images.
Author Response
REVIEWER 1
We read with interest comprehensive opinion of the Reviewer 1. The Reviewer comments are presented in 8 points.
- We don’t agree with statement that “PNI of cervical cancers is associated with LVS and LNM”. When we take into account the reports dealing with PNI in cervical cancer the above fact is confirmed only in some of them. In some papers, the author didn’t find any association and in many it is not studied. We agree that PNI status is not stated in current cervical cancer treatment guidelines. We also agree that “PNI is associated with the depth of the invasion, tumor size and more advanced stage of the disease in early-stage cervical cancer with high-risk of recurrence”, but it was taken into account in the algorithm suggested by us.
- The Reviewer 1 described the recommendations for NSRH very generally. The Reviewer 1 states that “NSRH is recommended for selected cases under the condition that the NSRH does not impair the radicality of surgery”. Meanwhile, radicality is also removing involved nerves. The reviewer 1 asks us to perform the study of PNI status which would permit to assess imaging method sensitivity and specificity. In the era when the incidence of cervical cancer is diminishing this study will be prolonged and time consuming - recruiting the huge number of cervical cancer patients, application of imaging, CRH, pathological assessment of PNI in post operative pathological examinations. Thus we agree with reviewer 1 who “does not believe” that it is possible.
- De facto we suggest that in future two kinds of biopsy may be applicable. First is intraoperative frozen section or preoperative laparoscopic biopsy. We want to stress that biopsy of uterine branch of inferior hypogastric plexus and has not any serious side effects parallel to hysterectomy.
- In the figure we presented the biopsy on right side, but it is not obligatory. We agree with the Reviewer 1 that cone biopsy “might be a diagnostic value”, and we treat it as a stage in diagnostic work up and we do not exclude it. It should be stressed that FIGO 2018 permitted applying imaging method to assess the stromal invasion in cervical cancer.
- The report presents a highly novel idea. If this idea was fully developed and included in NCCN guidelines, it would not be new and not worth considering. Additionally, due to above mentioned reasons, the original analysis on this future system is not available and premature.
- We are grateful for this precious remark which helped to improve our report. We changed the ureter in legend to Figure 1.
- The proposed by us very preliminary algorithm is a trial of an answer to surgeon who hesitates before choosing NSRH or CRH, and it is not the final accepted version of the procedure. This requires further research.
- In the previous mail the Editor asked us to describe our report type as “perspective”. We fully accepted it.
Finally, we would like to thank very much to Reviewer 1 for the in-depth analysis and we hope that our answers will be, at least partly, satisfactory.
Reviewer 2 Report
This study reviewed perineural invasion and novel algorithm during nerve-sparing radical hysterectomy.
-
The authors showed laparoscopic biopsy of uterine branch of inferior hypogastric plexus. According to frozen biopsy of uterine branch of inferior hypogastric plexus, nerve-sparing or conventional radical hysterectomy was performed. Only small part of uterine branch of inferior hypogastric plexus was evaluated as frozen biopsy. So, how did you evaluate perineural invasion of uterine branch of inferior hypogastric plexus exactly. Small portion of neural portion may not represent perineural invasion status exactly.
-
The exact prognostic value of perineural invasion was not fully understood in cervical cancer. Did you think perineural invasion was intermediate or high risk factor? If perineural invasion was considered as high risk factor, algorithm may be changed.
Author Response
REVIEWER 2
We appreciate the opinion of Reviewer 2 dealing with our report. The Reviewer 2 has concentrated on two questions. The first one is random laparoscopic biopsy of the uterine branch of inferior hypogastric plexus. We fully agree with the notice that during it only small part uterine branch is taken and evaluated. But it’s obvious that during random not guided, biopsy only small part of tissues is obtained. Additionally, as in other circumstances if guiding methods are unavailable the random biopsy remains the only option. We also agree with the Reviewer that suggested by us biopsy may not be representative in the assessment of PNI status. But firstly: the uterine branch of inferior hypogastric plexus is nervous tissue which can be sacrificed because parallel hysterectomy permits it. Secondly, biopsies of other pelvic nervous system are impossible although it would permit to assess the range of PNI.
We agree with statement of the Reviewer 2 that “The exact prognostic value of perineural invasion was not fully understood in cervical cancer”, but description of it is not the subject of our report. We only tried to build an algorithm of preoperative diagnosis of PNI useful for choosing between the CRH or NSRH in patient qualified to RH.
Reviewer 3 Report
This is a highly novel research proposing a new algorithm for the surgical approach of cervical cancer with perineural invasion based on the results of previous studies. There is no problem with the logical development of the manuscript, and it is considered worthy of acceptance without revision.
Author Response
We are very thankful to Reviewer 3 for finding our report highly novel based on the results of previous studies. We suspect the Reviewer 3 had no objections because he realized that this highly novel concept is not based on fully documented studies.
Round 2
Reviewer 1 Report
Since the authors made a big effort to respond to my question, and only one change at one site was made, I cannot add any comment to it. I suggest that part of their explanation could be put into the article.
Author Response
Thank you for your comments, we have made change to the manuscript market in red.
Reviewer 2 Report
Thank you for your efforts to revise your manuscript as reviwers' comments.
Adding the author respose of reviewer comments to your manuscipt may improve your study.
- "We fully agree with the notice that during it only small part uterine branch is taken and evaluated. But it’s obvious that during random not guided, biopsy only small part of tissues is obtained. Additionally, as in other circumstances if guiding methods are unavailable the random biopsy remains the only option. We also agree with the Reviewer that suggested by us biopsy may not be representative in the assessment of PNI status. But firstly: the uterine branch of inferior hypogastric plexus is nervous tissue which can be sacrificed because parallel hysterectomy permits it. Secondly, biopsies of other pelvic nervous system are impossible although it would permit to assess the range of PNI."
- "The exact prognostic value of perineural invasion was not fully understood in cervical cancer”, but description of it is not the subject of our report. We only tried to build an algorithm of preoperative diagnosis of PNI useful for choosing between the CRH or NSRH in patient qualified to RH."
Author Response

(The authors gave the same response as above.)
